# Research on Multimodal Fusion of Temporal Electronic Medical Records

**DOI:** 10.3390/bioengineering11010094

**Published:** 2024-01-18

**Authors:** Moxuan Ma, Muyu Wang, Binyu Gao, Yichen Li, Jun Huang, Hui Chen

**Affiliations:** 1School of Biomedical Engineering, Capital Medical University, No. 10, Xitoutiao, You An Men, Fengtai District, Beijing 100069, China; mmx0328@mail.ccmu.edu.cn (M.M.);; 2Beijing Key Laboratory of Fundamental Research on Biomechanics in Clinical Application, Capital Medical University, No. 10, Xitoutiao, You An Men, Fengtai District, Beijing 100069, China

**Keywords:** multimodal fusion, time-series electronic medical records, prediction, multimodal adaptation gate, attention backtracking

## Abstract

The surge in deep learning-driven EMR research has centered on harnessing diverse data forms. Yet, the amalgamation of diverse modalities within time series data remains an underexplored realm. This study probes a multimodal fusion approach, merging temporal and non-temporal clinical notes along with tabular data. We leveraged data from 1271 myocardial infarction and 6450 stroke inpatients at a Beijing tertiary hospital. Our dataset encompassed static, and time series note data, coupled with static and time series table data. The temporal data underwent a preprocessing phase, padding to a 30-day interval, and segmenting into 3-day sub-sequences. These were fed into a long short-term memory (LSTM) network for sub-sequence representation. Multimodal attention gates were implemented for both static and temporal subsequence representations, culminating in fused representations. An attention-backtracking module was introduced for the latter, adept at capturing enduring dependencies in temporal fused representations. The concatenated results were channeled into an LSTM to yield the ultimate fused representation. Initially, two note modalities were designated as primary modes, and subsequently, the proposed fusion model was compared with comparative models including recent models such as Crossformer. The proposed model consistently exhibited superior predictive prowess in both tasks. Removing the attention-backtracking module led to performance decline. The proposed model consistently shows excellent predictive capabilities in both tasks. The proposed method not only effectively integrates data from the four modalities, but also has a good understanding of how to handle irregular time series data and lengthy clinical texts. An effective method is provided, which is expected to be more widely used in multimodal medical data representation.

## 1. Introduction

With the vigorous development of internet information technology, data management and transmission have become more efficient, and medical institutions have also built many electronic medical records (EMRs) information databases. Among the many modern medical data, electronic medical record data are one of the most important medical data resources [1,2,3]. Particularly noteworthy is the growing adoption of EMR systems in numerous hospitals, owing to the improved implementation of global hospital information systems [4].

Unlike data from clinical trials or other forms of biomedical research, EMR data are not generated for a specific purpose, the primary purpose of employing EMR systems is to record the health status of patients. It encompasses a multitude of data from various sources and of different natures [1,5,6], including medical history, diagnostic test results, medication usage, and demographic information [7]. Characterized by its diversity and complexity. This makes the effective extraction of patient features and the establishment of a multimodal representation from the raw EMR data a challenging task.

In the medical domain, research employing multimodal data has exhibited notable diagnostic proficiency, thereby aiding the enhancement of healthcare practices and cost reduction [8,9]. Nevertheless, prior investigations concerning EMRs have often been confined to predictive modeling employing structured tabular data or unstructured clinical notes [10,11,12] alone. For instance, some studies have employed clinical notes to predict in-hospital patient mortality and disease outcomes [10], utilized international classification of disease (ICD) codes for patient disease prediction, and similarity clustering analysis [13], or employed temporal text to forecast congestive heart failure and chronic obstructive pulmonary disease [14]. While these studies have achieved satisfactory results, in practice, clinical experts often evaluate patients based on information from multiple sources rather than a single modality of data. Research has shown that deep learning models based on multimodal medical data have higher predictive performance than using a single modality, providing valuable insights for clinical decision making [8].

Furthermore, EMR data not only encompasses data from multiple modalities but also reflects the static and temporal health status of patients during their hospitalization. Among these, the temporal features of a patient encapsulate the progression of their condition and the course of treatment, providing vertical information into their health. This information aids clinical practitioners in making more precise diagnoses and treatment decisions. In recent years, deep learning, endowed with potent modeling and generalization capabilities, has garnered significant attention as an emerging approach for handling temporal data. However, due to the sparsity and varying lengths of medical temporal data, commonly used deep learning methods such as recurrent neural network (RNN) and long short-term memory (LSTM) often struggle to effectively represent the temporal features of patients.

Therefore, various relevant studies have proposed solutions: Lee et al. extracted continuous vital sign parameters measured within the first 24 h after a patient’s admission to the intensive care unit (ICU), calculating the maximum and minimum values to predict the risk of post-discharge mortality [15]. Wang et al. transformed the original time series of vital signs and laboratory measurement data into usable hourly features through interpolation and organization, aiming to reduce missing data and enhance robustness [16]. Although both methods have achieved satisfactory results, they still have some limitations. Both only predict data within a few hours after entering the ICU, while the patient’s condition during hospitalization also plays a crucial role in predicting subsequent tasks. Additionally, for the former, although the model’s predictive performance is improved compared to models using only static data, selecting extremes within a short period for time series data may lead to the loss of essential information. As for the latter, segmenting data into hourly intervals results in a loss of granularity present in the original data. Furthermore, both approaches overlook clinical records, especially those in the form of time series data with significant clinical value [17,18]. Currently, the integration of temporal information with textual data are becoming a popular choice to enhance the performance of prediction tasks. Common approaches include using RNN and its variants or architectures based on transformers [19]. However, for time series medical data, the uneven distribution of data and the longer length of clinical texts pose challenges in employing conventional models like transformers. Thus, there is a need for a class of models that can handle both time-series medical data and lengthy medical texts simultaneously.

Cardiovascular disease is one of the most threatening health conditions [20]. Global statistics show that cardiovascular diseases account for one third of deaths every year, with acute myocardial infarction (AMI) having the highest morbidity and mortality. Stroke is a serious disease with high morbidity, mortality, and disability rates [21]. Especially in China, where 1.8 million people died of stroke in 2016 [22]. Therefore, using AMI and stroke patient data for tasks such as in-hospital mortality prediction is crucial in clinical practice. This predictive approach plays a key role in enabling medical professionals to enhance patient insights, produce timely tailored treatment plans, and reduce healthcare-related costs [23,24,25].

Research indicates that incorporating patients’ time series data into predictive models can improve the model’s predictive performance [26]. Therefore, how to effectively integrate heterogeneous data types, including time series and static data, as well as text and tabular data, has become a new challenge in EMR modeling. This study aims to devise a novel multimodal fusion model, termed the time series multimodal adaptation gate (T-MAG). This model employs four encoders to handle four types of patient EMR data, specifically: temporal tabular information, temporal notes, static tabular information, and static notes. The multimodal adaptation gate (MAG) [27] is then employed to effectuate multimodal fusion, with the addition of an attention-backtracking module to enhance the model’s capacity to address long-term dependencies. An overview of the multimodal fusion model we propose, and its application, are depicted in Figure 1.

The contributions of this paper can be summarized as follows:We propose an integrative approach that combines time series clinical note data, time series tabular data, static clinical note data, and static tabular data, resulting in improved performance on two types of predictive tasks;Addressing the irregularity and non-uniformity of medical time series data, we employ a time window to mitigate these challenges. Simultaneously, the integration of an attention-backtracking module enhances our model’s ability to capture long-term dependencies;By comparing two types of prediction models, utilizing LSTM and a deep neural network (DNN), we demonstrate that neglecting the temporal sequence information embedded in time series data can have detrimental effects on the predictive performance of the model.

The remainder of this paper is organized as follows. Section 2 first introduces the data used in the article, followed by a detailed description of the proposed time series multimodal fusion method. Finally, the experimental and comparative methods used in this study are presented, and the results are presented in Section 3. Section 4 discusses the strengths and limitations of LSTM. The Appendix A provides information on how the time window size used in this study was determined.

## 2. Materials and Methods

### 2.1. Dataset and Data Preprocessing

In this study, we employed a proprietary dataset sourced from the EMR system of a tertiary hospital in Beijing, China, covering the period from January 2014 to December 2016. The data utilized in this study encompass patient demographic features, medications, laboratory tests, and clinical notes, comprising admission notes, progress notes, and discharge summaries. The personal information of patients (such as names, ID numbers, addresses, and phone numbers) has been entirely removed from the dataset to ensure the data are used in an anonymous and secure manner. The ethical approval for this research was granted by the hospital’s Institutional Review Board.

The patient data we utilized consist of two types: tabular data and textual data. Each data type can be further categorized into temporal and static data based on whether they vary over time, resulting in a total of four data subsets. In this paper, we refer to each of these data subsets as a kind of modality. Please refer to Table 1 for detailed introduction of each data.

The introduction of each modality is as follows:Static tabular data: This is obtained from the admission records and includes 6 demographic attributes of the patient, as well as 7 basic physical examination parameters;Static note data: Also obtained from admission records, this category encompasses 15 types of information, including the patient’s complaints, medical history, specialized examinations, admission diagnoses, and confirmed diagnoses. Additionally, due to the absence of punctuation in some text data, such as in-patient diagnoses, we utilized the Jieba segmentation tool for tokenization [28]. This tool is a specialized Chinese tokenization tool that automatically identifies new words and proper nouns based on Chinese text;Temporal tabular data: Extracted from progress notes, this contains laboratory test results and vital signs measured at two or more time points during the hospital stay. It includes a total of 95 parameters;Temporal note data: Derived from progress notes, this category consists of daily ward round records for each day of the patient’s hospitalization. In cases where multiple rounds occur on the same day, only the first-round record of the day is selected.

The two types of static data generally remain constant during the hospital stay, whereas the two types of temporal data are subject to updates as time progresses.

For the two types of time series data, dealing with them directly using traditional algorithms poses a challenge due to the varying lengths of patient hospital stays and the uneven distribution of time series data generated during hospitalization. Therefore, we conducted the following preprocessing steps for the patient’s time-series information: First, given the substantial presence of missing values in time series data, we employed an interpolation method to fill in the gaps. Then, since patients’ lengths of stay varied significantly, and only a small fraction of patients had stays exceeding 30 days, we merged the data from stays exceeding 30 days with the data from the 30th day. We then padded the data for patients with a hospital stay of less than 30 days to extend it to a uniform length. To mitigate the impact of varying lengths of hospital stays, we use rolling time windows to decompose time series note data and time series table data into n subsequences of equal size, where n is the number of subsequences. In this study, n is set to 10. For details regarding the setting of the time window size, please refer to Appendix A.

For the AMI dataset, we selected 1639 records corresponding to patients diagnosed with AMI. The diagnosis of AMI was confirmed through ICD-10 codes I21 and I22. Subsequently, we excluded records with hospitalization durations of less than 1 day and records lacking clinical notes or laboratory test results. This process resulted in a total of 1271 records meeting the specified criteria. Among these, 91 records (7.16%) had in-hospital mortality outcomes, and 1029 records (81%) had hospitalization durations within two weeks. For the Stroke dataset, the final number of records meeting the criteria was 6450. The diagnosis was confirmed through ICD-10 codes I60~I64, I66, and I67.8. Among these records, 107 (1.66%) had in-hospital mortality outcomes, and 5679 records (88%) had hospitalization durations within two weeks.

### 2.2. The T-MAG Model

The fusion architecture proposed in this study, as illustrated in Figure 2, is divided into two main components: representation and fusion. Initially, each of the four data modalities is independently subjected to representation. Subsequently, they are individually fed into static and temporal MAG for fusion, thereby yielding the static fusion representation and temporal fusion representation. After concatenation, these representations are input to an LSTM module to derive the ultimate fused representation for predictive purposes. In the following sections, we will provide a detailed description of each of these components.

#### 2.2.1. Representation for Each Modality

In this section, we elaborate on patient representation for the four modalities used in our research. For the four modalities of data, we employ distinct encoders to represent each of them:Feature embedding for static tabular data

Our static tabular data are divided into two categories: continuous variables (such as age, height, and weight) and discrete variables (such as gender and ethnicity). The former needs to be discretized based on their respective reference standards; for example, categorizing body temperature as high, normal, or low based on the normal range. Subsequently, we employed one-hot encoding to represent the data, generating the static tabular representation, where each feature in the vector represents a unique dimension. For static tabular data $I_{st}$, the computation method of static tabular representation $F_{st}$ is as follows:(1)$$F_{st}=Onehot(I_{st})$$

Feature Embedding for static note data

For the static note data, due to its relatively simple structure, we utilized a word embedding technique like doc2vec on tokenized input to obtain the representation of the patient’s static notes, the hidden layer dimension of Doc2Vec was set to 300. For static note data $I_{sn}$, the computation method of static note representation $F_{sn}$ is as follows:(2)$$F_{sn}=doc2vec(I_{st})$$

Feature Embedding for temporal tabular data

For the time series tabular data, we began by processing each subsequence of the time series tables separately. Initially, we employed a fully connected neural network with a rectified linear unit (ReLU) activation to encode each subsequence individually. Subsequently, we fed each encoded subsequence into an LSTM neural network to obtain the time-structured representation for each subsequence.

For time series tabular data $I_{tt}=(I_{tt}^{1},I_{tt}^{2},I_{tt}^{3}\ldots I_{tt}^{i}\ldots I_{tt}^{n})$, we initially processed each subsequence separately. Initially, we employed a fully connected neural network with ReLU activation to independently encode each subsequence. Subsequently, we input each encoded subsequence into a 2-layer LSTM to obtain the representation for each subsequence. The computation method for the representation of each time series tabular data $F_{tt}^{i}$ is as follows:(3)$${F_{tt}^{i}}^{\prime }=ReLU\left(Linear\left(I_{tt}^{i}\right)\right)\;F_{tt}^{i}=LSTM({F_{tt}^{i}}^{\prime })$$

Feature Embedding for temporal note data

Common methods for processing clinical records involve natural language processing models such as Transformer and bidirectional encoder representations from transformers (BERT). However, these methods impose certain limitations on the length of each input text. When dealing with texts exceeding a fixed length, they are often split into multiple segments for separate processing. This not only results in potentially incomplete semantics in segmented portions but also makes it challenging to capture long-term dependencies. Given the characteristics of lengthy and voluminous content in the time series clinical note data $I_{tn}=(I_{tn}^{1},I_{tn}^{2},I_{tn}^{3}\ldots I_{tn}^{i}\ldots I_{tn}^{n})$, in this study, we utilized Transformer-XL [29] for encoding, which is a variant of the Transformer architecture, In this study, the number of layers for Transformer-XL is set to 12. Transformer-XL introduces improvements such as a recurrence mechanism and positional encoding, enabling more efficient processing of lengthy annotations and adeptly capturing long-term dependencies. Subsequently, we input each encoded subsequence into a 2-layer LSTM neural network to obtain the temporal structure information for each subsequence. The computation method for the representation of each time series clinical note is as follows:(4)$${F_{tn}^{i}}^{\prime }=TransformerXL(I_{tn}^{i})\;F_{tn}^{i}=LSTM({F_{tn}^{i}}^{\prime })$$

#### 2.2.2. Fusion for Each Modality

The concept of a multimodal adaptation gate was initially introduced by Wang et al. in 2019 [30], primarily in the context of multimodal sentiment analysis, and later extended to the medical domain. The core idea involves treating one modality as the primary modality and the others as auxiliary modalities. The influence of auxiliary modalities on the primary modality is represented as a displacement vector H. The fusion of multiple modalities is interpreted as the addition of the primary modality and the displacement vector. The MAG constructs specific modality attention gates to simulate the influence strength of other modalities on the primary modality, thereby controlling the importance of auxiliary modality embeddings. This not only allows for comprehensive perception of information across different modalities but also effectively removes redundant and irrelevant information from the representation through gating mechanisms. The data from various modalities in the EMR have varying degrees of importance in assessing the patient’s condition, and there is redundancy between modalities. Therefore, we employ the MAG to integrate data from the four modalities.

In this study, we utilized four modal representations; however, the direct fusion of time series and static representations can obscure the inherent temporal sequence information of the former. Hence, we performed separate fusions for the two categories: static representations and time series representations, resulting in fused static representations and fused time series representations. In the initial application of MAG in the field of sentiment analysis, the choice of the primary modality is evident because text consistently contains, in relative terms, the richest emotional information. However, in the medical domain, this is harder to determine as each modality of data can provide its own unique information. So, in the above formula, two types of notes are selected as the main modality, while the selection of the primary modality will be further discussed in subsequent chapters. The representations of the two modalities are fused separately through their corresponding MAG. Here, the fusion of the static representation is taken as an example. Similar formulas can also be used for the fusion of the time series representation. The static clinical note $F_{sn}$ is chosen as the primary modality, while the static tabular $F_{st}$ serves as the auxiliary modality.

The static MAG is calculated as follows:(5)$$g_{S}=\sigma (W_{gS}\left[F_{Sn};F_{St}\right]+b_{gS})$$
where $g_{S}$ is the gating value of static tabular modality $W_{gS}$ is the weight matric and $b_{gS}$ is the scalar base, and σ(x) is the sigmoid function.

The displacement vector $H_{S}$ is calculated by $F_{St}$ multiplied by its gating values:(6)$$H_{S}=g_{S}\cdot \left(W_{hS}F_{St}\right)+b_{hS}$$
where $W_{hS}$ is the weight matric for static tabular modality and $b_{hS}$ is the bias vector.

Then, a weighted summation is performed between the main feature $F_{Sn}$ and the displacement vector $H_{S}$ to create a multimodal representation $E_{S}$:(7)$$E_{S}=F_{Sn}+\alpha_{S}H_{S}\;\alpha_{S}=min(\frac{{\left|\left|F_{Sn}\right|\right|}_{2}}{{\left|\left|H_{S}\right|\right|}_{2}}\beta_{S},\;1)$$
where $\beta_{S}$ is a randomly initialized hyper-parameter training with the model. ${\left|\left|F_{Sn}\right|\right|}_{2}$ and ${\left|\left|H_{S}\right|\right|}_{2}$ are the $L_{2}$ norm of $F_{Sn}$ and $H_{S}$. The scaling factor $\alpha_{S}$ is used to restrict the effect of the displacement vector H to a desirable range.

For T-MAG, we introduce an additional attention-backtracking module to enhance the model’s ability to capture long-term dependencies in temporal representations. We employ a self-attention mechanism that utilizes information from previously represented subsequences to complement the representation of the current subsequence. The purpose of this is to strengthen the model’s capability to handle long-term dependencies within the data. Specifically, for the representation $F_{t}^{i}$ of each subsequence, we calculate attention weights between $F_{t}^{i}$ and the representations of previous subsequences ($F_{t}^{1}\sim F_{t}^{i-1}$) using the self-attention mechanism. We then perform a weighted sum with each original representation to complement $F_{t}^{i}$, generating a new subsequence representation Q. This new representation not only incorporates information from the current time but also includes crucial information from past times, enhancing the model’s capacity to handle long-term dependencies.

After obtaining the two fusion representations separately, we treat the static fusion representation as the representation for day 0, concatenating it before each time series subsequence fusion representation. These concatenated representations are then input into a 2-layer LSTM to obtain the final patient fusion representation, denoted as Z.

### 2.3. Evaluation of the T-MAG-Based Multimodal Fusion Model

#### 2.3.1. Tasks and Indexes for the Performance Evaluation

The performance evaluation of the fusion model typically relies on the performance of the classification model based on the fused representations. Therefore, we utilized the following two classification tasks to assess the performance of our model: predicting in-hospital mortality and determining whether a patient is likely to have an extended length of stay. In the latter, 0 indicates a hospital stay of 14 days or less.

To assess these two tasks, we employed an F1 score, area under the receiver operating characteristic curve (AUROC), and area under the precision-recall curve (AUPRC) for performance evaluation, with AUROC being the primary metric. F1 and AUPRC are utilized to provide insights into the performance of AUROC on imbalanced datasets [26,31,32].

For the two binary classification prediction tasks used in this study, we employ a sigmoid layer for prediction, with the following formula:(8)$$\overset{ˇ}{Y}=sigmoid(Linear(Z))$$

Following research by Tang [5], the loss function used in this paper is:(9)$$L=-\frac{1}{B}\sum_{i=1}^{B}Y_{i}\log\left(\overset{ˇ}{Y_{i}}\right)+\left(1-Y_{i}\right)\log\left(1-\overset{ˇ}{Y_{i}}\right)$$
where *Y* is the true value and $\overset{ˇ}{Y}$ is the predicted value. We use the cross-entropy between the true value and the prediction to arrive at the loss.

#### 2.3.2. Evaluation Experiments

Firstly, each MAG needs to select a primary modality, which plays a leading role in the fusion process. To investigate the impact of different primary modality selection strategies on the model’s prediction results, we constructed four different T-MAG models. We conducted predictions on two clinical tasks, respectively, to explore the optimal combination of primary modalities in different tasks.

Subsequently, to investigate whether adding different modalities positively affects the model’s performance, we conducted analyses using various subsets derived from the complete dataset and employed in-hospital mortality prediction as the designated forecasting task. Specifically, we conducted our research using four data subsets, each containing only two of the four original modalities. We selected four subsets to compare the influence of each modality on the model performance: “only static” using only static tabular data and static notes, “only time-series” using only time series tabular data and time series notes, “only tabular” using only static tabular data and time series tabular data, and “only notes” using only static notes and time series notes. By comparing the results of these subsets, we aimed to identify the individual contributions of the four modalities to the predictions.

Finally, we compared the obtained optimal choices with a series of Compare models, whose performance has been validated in prior studies. The Compare models used will be introduced subsequently.

#### 2.3.3. Comparative Models

We have selected the following 2 categories of models as Compare models: traditional machine learning methods, deep learning methods, and multimodal fusion methods.

Deep learning methods: In recent years, deep learning techniques have achieved significant success in various domains by constructing deep hierarchical features and effectively capturing long-range dependencies in the data [19]. Therefore, we have employed two neural network architectures—deep neural network (DNN) and long short-term memory (LSTM). For DNN, since it is difficult to process time series data, we splice all static representations before the time series representation and ignore the time series information contained in it. For LSTM, we first splice the table representation and the note representation, respectively, and then define the static fusion representation as the representation on the 0th day of admission.

Fusion methods: We also introduced several recently proposed multimodal fusion models to compare the effectiveness of our fusion approach, categorizing them into the following types:Models using a simple fusion method—fusion-convolutional neural network (Fusion-CNN) and fusion-LSTM [33];Transformer-based fusion models—Multimodal Transformer (MulT) [34], Crossformer [35], and patch time series transformer (PatchTST) [36];Models applied to time series EMR—Multivariate irregularly sampled time series (MISTS)-fusion [37] and Glaucoma-fusion [38];MAG-based fusion models—MAG-LSTM and MAG-DNN.

These multimodal fusion models all divide data into three modalities: static table data, time series table data, and clinical notes. Clinical notes are not divided into time series and static categories. For Fusion-CNN and Fusion-LSTM, these two models obtain the representations of the three modalities, respectively, and then directly connect them to obtain the final fusion representation. The former adopts 2-layer CNN and max-pooling to represent temporal data, while the latter uses bi-directional long short-term memory (BiLSTM) to process temporal data, which can effectively capture long-term dependencies in sequence data. In the MulT model, each modality is encoded individually, and then cross-modal attention is employed to pairwise fuse the representations of the three modalities. Subsequently, it utilizes self-attention to obtain a fused representation of the three modalities. This fused representation is then concatenated to produce the final representation. Crossformer and PatchTST are new models proposed in the past year based on Transformer. The former focuses on the model’s capture of cross-modal dependencies of different modalities. It first embeds the time series data segmentally and then uses two attention mechanisms to capture cross-time and cross-modal dependencies. Modal dependencies, and finally uses a hierarchical encoder–decoder to fuse multimodal data. The latter divides the time series into patches. Each patch contains information on multiple time steps, and the sequence of each variable is independently mapped to an independent embedding for modeling. Meanwhile, we also considered two medical multimodal fusion models released in 2023: MISTS-fusion and Glaucoma-fusion. MISTS-fusion is proposed to handle irregularities in tabular data and clinical notes. This method employs a gating mechanism to dynamically fuse imputed embeddings and a time attention module to obtain interpolated representations for time series tabular data. Simultaneously, the TextEncoder encodes time series notes, which are then input into the time attention module to acquire interpolated representations for time series notes. Finally, a cross-modal attention mechanism is used for multimodal fusion. In contrast, Glaucoma-fusion predicts whether patients will undergo glaucoma surgery in the next 12 months using clinical notes and tabular data from the preceding 4–12 months. For these two types of data, glaucoma-fusion uses CNN and XGBoost for representation, followed by concatenation fusion. MAG-LSTM and MAG-DNN are based on a study by Yang et al. [28], where time series annotations were chosen as the main modality. They used MAG to integrate three modes: static representation, time series list representation, and time series annotation representation to obtain the final fused representation. The main difference between them is the neural network architecture used to represent time series list data and time series note data.

#### 2.3.4. Ablation Experiments

Long-term dependencies refer to the phenomenon in time-series data where the current state or event may be influenced by events or states that occurred significantly earlier in time. In medical data, since a patient’s condition is a developing process, capturing long-term dependencies in the patient’s time-series data allows for better tracking of the progression of their condition. It also enables predictions and warnings about the patient’s condition based on past data. In this study, an attention-backtracking module was employed to capture these long-term dependencies. To investigate its impact on the model’s predictive performance, we removed the attention-backtracking component from the model and compared the results obtained with the original model across two clinical tasks.

#### 2.3.5. Experimental Setup

The traditional machine learning methods (i.e., LR and RF) used in this study were implemented using scikit-learn [31]. The deep learning models were implemented using PyTorch [39]. All deep learning models were trained using the Adam optimizer, with a learning rate of 0.0001, a batch size of 64, and a maximum epoch count set to 50. The dataset was randomly partitioned into training, validation, and test sets in an 8:1:1 ratio. For both tasks, the primary evaluation metric was AUROC. Additionally, we reported F1 score and AUPRC to provide insight into the interpretation of AUROC on imbalanced datasets. For each model, we selected the parameter set that yielded the minimum validation loss. All models were implemented using PyTorch 1.10 and trained on a workstation equipped with an Intel Xeon Gold 5218 processor, 512 GB RAM, and a 16 GB NVIDIA Tesla T4 GPU.

## 3. Results

In this section, we present the results of comparative and ablation study. Each reported performance metric is the average score of five runs with different data splits.

### 3.1. Impact of the Main Modality on the Model Performance

Firstly, we compared four main modality selection strategies on two tasks, and the experimental results are shown in Table 2. It is evident that selecting two types of note modes as the main modality yields the best results for both tasks (AMI: AUROCs 0.928 and 0.881, Stroke: AUROCs 0.954 and 0.847), which also demonstrates the high importance of clinical records. As a result, in subsequent experiments, we will consistently use two types of note modes as the main modality to achieve enhanced predictive performance.

### 3.2. Impact of Different Subsets on Model Performance

The corresponding results for the subset are shown in Figure 3. From the results, using the complete data set always yields better results than using partial data subsets, with AUROC being at least 0.08 and 0.007 higher in two types of data sets than in the other four types. Among the four subsets, the best performer is “only note”, whose AUROC is 0.848 and 0.938 in the AMI dataset and the Stroke dataset, respectively. The “static only” subset performed the worst (AUROCs 0.752 and 0.879), probably because it does not contain any time series data, especially the time series annotations, which contain the most information.

### 3.3. Results of Comparative Experiments

Subsequently, we compared the best choice with the Compare model. In the two data sets, the performance of various models on in-hospital mortality prediction and long-term hospitalization prediction tasks are shown in Table 3 and Table 4, respectively. It is obvious that our T-MAG model achieved the best prediction results in both tasks, in AUROC (AMI: 0.928 and 0.881, stroke: 0.954 and 0.847), AUPRC (AMI: 0.363 and 0.632, stroke: 0.455 and 0.508), and F1-score (AMI: 0.535 and 0.478, stroke: 0.671 and 0.376), exceeding the Compare model. In addition, the prediction performance of various fusion models in the comparative experiments is generally better than the other two types of models. This demonstrates the effectiveness of employing multimodal fusion techniques to supplement information from different modalities to obtain a more comprehensive understanding of a patient’s condition. Among the fusion models, the three models using MAG as the fusion method outperformed most other models, highlighting the competitiveness of utilizing MAG as the fusion method.

### 3.4. Results of Ablation Experiments

We investigated the impact of incorporating an attention-backtracking component on the model’s performance. In the AMI dataset, removing this module resulted in an AUROC of 0.889 for the in-hospital mortality task and 0.837 for the long-term hospitalization duration task. Meanwhile, in the Stroke dataset, the AUROCs for the two tasks were 0.928 and 0.772, respectively. From these results, it can be inferred that eliminating the attention-backtracking component leads to a decline in predictive performance.

## 4. Discussion

With the widespread adoption of EMR systems, an increasing number of researchers are turning to deep learning methods for medical studies, but most of these studies are limited to single-mode electronic medical records. Typically, research focuses on the fusion of imaging data, tabular data, and imaging reports [8,40], while research on time series clinical notes like progress records are relatively scarce. These data encompass information about the patient’s condition progression, crucial auxiliary test results, and clinical expert opinions, containing rich temporal information. Furthermore, the time-series tabular data include multiple measurements of laboratory indicators and vital signs during the hospitalization, serving as the basis for clinical doctors to assess the patient’s condition [9,10,41,42]. To fully leverage these valuable data, we devised an innovative multimodal fusion model capable of integrating data from different modalities. Our approach not only effectively integrates information from various modes but also adeptly captures the intricate temporal dependencies in time series data. The results of two medical prediction tasks emphasize the superiority of our fusion model over other advanced fusion methods. Time series data provide information about the changing trends of patients over a period, reflects the development of the disease course, and can reveal the dynamic changes in the disease. Static data usually include the patient’s basic information, physiological indicators, and other basic files, as well as the patient’s historical medical information. The fusion of these two types of data can provide a more comprehensive view of the patient and help doctors fully understand the patient’s condition and health situation.

In the medical multimodal field, fusion methods based on attention mechanisms are widely used [43], such as the use of multi-channel attention fusion in RAIM [44], which integrates continuous patient monitoring data, and the extensive use of transformers in other domains [45,46]. The attention mechanism allows the model to dynamically adjust the degree of attention to different modal information. This flexibility enables the model to better adapt to the characteristics of different tasks or different samples, thereby improving the generalization performance of the model. Yang et al. took the lead in introducing the multimodal fusion MAG method [27] into the medical field and proved its effectiveness on multimodal EMR data. Compared with other fusion methods, fusion using MAG can adapt to different task requirements and data characteristics by adjusting the selection of main modalities, and the gating mechanism contained in it can effectively remove redundancy in the data. However, existing research using MAG for fusion has not paid enough attention to time series data and only treated them on the same level as static data.

Among the other comparative models, the two with the best predictive performance are MISTS-fusion and Crossformer. The former addresses irregular intervals in medical time series data by handling each modality dataset separately through modules like imputation and time attention, followed by multilayered cross-modal attention fusion. However, this study only utilizes data within the first 48 h of admission in the original dataset, neglecting long-term dependencies and thus showcasing more prominent effects on the in-hospital mortality task. The latter, as in our study, segments time series into fragments for representation and incorporates two-stage attention to capture long-term dependencies and interaction between patterns. However, it does not specifically address medical data. Due to its superior performance in capturing temporal information, especially for longer time series data, it performs better on tasks predicting extended hospitalization periods and closely approaches our model’s performance on the Stroke dataset. However, its limitation lies in the difficulty of maintaining ordered information in transformer-based models [47], and cross-dimensional attention interactions may introduce noise. Our proposed T-MAG fuses data from four modalities step by step. Initially, two MAGs were used to fuse two temporal data types, respectively, resulting in a preliminary fused representation of time series and static data. The static fused representation is then concatenated with the fused representation of each time series subsequence and fed into an LSTM to obtain the final fused representation. The use of MAG enables the model to adapt to different situations and achieve the best performance in both types of tasks. In addition, compared with the two comparison models (MAG-DNN and MAG-LSTM) that also use MAG, the time window reduces the impact of irregularities and missing data in time series medical data. At the same time, the added attention-backtracking module also makes our model more suitable thanks to its ability to capture long-term dependencies, which the original MAG does not have. Therefore, our model is better suited for predictions using medical time series data, particularly when there is a need to capture long-term dependencies in the data—advantages not present in other comparative models. As evident in both task categories, T-MAG demonstrates an improvement of at least 0.05 and 0.03 in AUROC compared to the other two models, validating the effectiveness of our enhancements. Among them, the models using LSTM (T-MAG and MAG-LSTM) are better than the models using DNN. This is especially true in the “long stay” task. This may be due to the “long-stay” mission’s emphasis on dynamic monitoring of patient status and treatment adjustments over an extended period. DNNs have difficulty retaining temporal information in input sequences, which may make it difficult to capture long-term trends. This verifies that retaining temporal information in the data has a positive impact on the performance of the model.

Time series data directly capture the dynamic health trends and treatment outcomes of patients [15,48]. Scholars have proposed that incorporating time series data into prediction models can enhance their performance [27,41,48]. Currently, RNN and their variants are commonly employed in handling medical time series data, as they are well-suited for sequential and time-related tasks [49,50]. Zhao et al. demonstrated that while RNNs and LSTM networks are adept at handling time series data, they do not possess long-term memory [51]. In the real world, most of the information unfolds over time. The human brain adeptly associates events stored in memory, capturing long-term dependencies even when events are temporally distant from each other [52]. The self-attention mechanism is designed based on this precursor, and substantial empirical evidence indicates its efficacy in enhancing learning and computation for long-term dependencies [53,54]. In our model, we employ an attention-backtracking module to capture long-term dependencies in medical time series data. Specifically, this module utilizes the self-attention mechanism to calculate the correlation between current time information and historical information, determining the significance of each position and generating the corresponding weighted representation. This module can disregard the actual time intervals between two sequential datasets, selectively choosing crucial portions from historical information, thereby enhancing the model’s ability to capture contextual information over longer time spans. This augmentation facilitates improved capture of critical trends and changes in continuous data, ultimately improving its predictive capabilities. In our model, we employ an attention-backtracking module to capture long-term dependencies in medical time-series data. This enhancement contributes to improved capture of key trends and changes in continuous data, ultimately improving predictive capabilities. In the study, the removal of the attention-backtracking module resulted in a decrease in model performance in both task categories, especially in the length of stay task. During hospitalization, patients may undergo various medical interventions, surgeries, medication treatments, etc. These complex medical processes may interact with each other, and accurate understanding of changes in the patient’s condition over a long time-range is essential for better predicting the patient’s future state. The comparison of the two datasets also validates this point, as the average length of hospital stay for patients in the Stroke dataset is higher than that in the AMI dataset, and the removal of attention-backtracking leads to a more significant performance decline.

However, our study still presents several limitations. Firstly, the sample size we employed is relatively limited, including EMR from only one hospital. This restricted sample size could potentially impact the performance of our models, introducing a risk of overfitting. Secondly, concerning temporal data, we employed relatively simplistic methods to handle issues related to missing data and capturing long-term dependencies. However, these methods may not fully capture the intricate temporal dynamics within the data. Further refinement of techniques for handling temporal information could lead to more accurate predictions. Lastly, our research is confined to just two specific clinical tasks—predicting in-hospital mortality and prolonged length of stay. While these tasks hold significant clinical importance, they do not provide a comprehensive assessment of our model’s capabilities. Future work could encompass a broader range of clinical tasks, such as predicting specific complications during hospitalization or engaging in multiclass disease diagnosis.

## 5. Conclusions

In this study, we have introduced a multimodal fusion model based on the MAG framework. This model cleverly integrates four different medical modalities and effectively captures the intrinsic long-term dependencies present in time series data. With the continuous advancement of technology and the emergence of more data sources, the integration of different data sources will play an increasingly important role in medical analysis. Our research provides a novel perspective on how to combine temporal and static multimodal data. Our research endeavors to advance the capabilities of deep learning models in the medical domain, opening new possibilities for stronger and more precise predictions. Our research can contribute to obtaining more comprehensive patient representations by integrating information from multiple modalities. Our model facilitates the deep reuse of medical big data, aiding healthcare professionals in making safer and more efficient diagnostic and treatment decisions. This, in turn, propels the advancement of clinical decision support and personalized medical research. In our future work, we plan to explore the applicability of the model in scenarios involving representations of other modalities of medical data, such as integrating clinical imaging or adapting the model for English-language datasets.

## Figures and Tables

**Figure 1 bioengineering-11-00094-f001:**
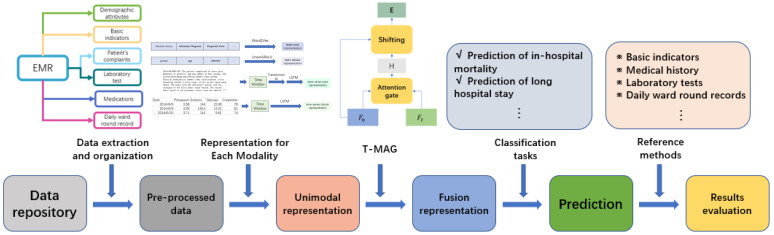
An overview of the study.

**Figure 2 bioengineering-11-00094-f002:**
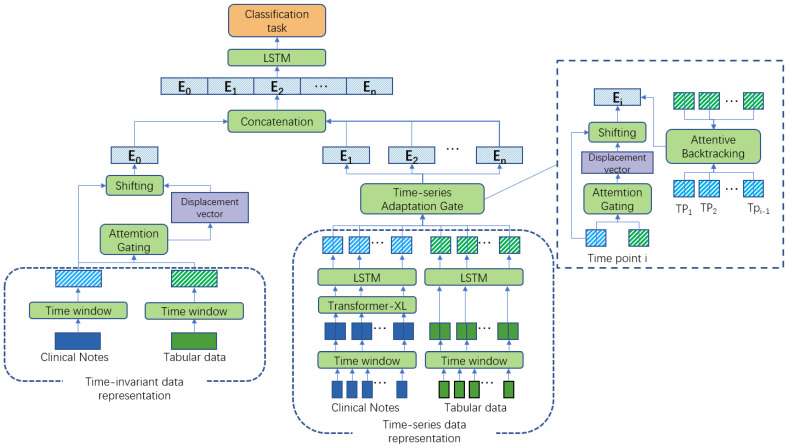
The overall architecture of T-MAG.

**Figure 3 bioengineering-11-00094-f003:**
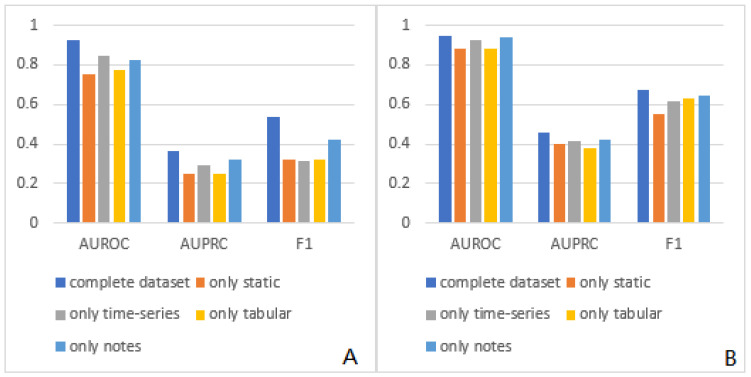
Prediction results of the complete data set and subsets in the AMI data set (**A**) and the Stroke data set (**B**).

**Table 1 bioengineering-11-00094-t001:** Details and examples of each modal data.

Modal	Feature Category	AMI Dataset	Stroke Dataset	Example
		Number of Features		
Static tabular	Gender	2	2	male, female
	Age	2	2	>60 years, ≤60 years
	Ethnicity	12	16	Han ethnicity, Hui ethnicity
	Marital status	3	3	married, divorced, unmarried
	Department	7	18	neurosurgery, vascular surgery
	Admission method	2	2	emergency department
	Basic indicators	7	7	height, weight, temperature
Static note	Chief Complaint	1	1	chief complaint
	Medical History	5	5	current medical history, past medical history
	Specialized Examination	2	2	specialist examination, auxiliary examination
	Admission Diagnosis	3	3	confirm diagnosis, supplementary diagnosis
	Characteristics	1	1	patient characteristics
	Diagnostic Basis	2	2	diagnostic basis, differential diagnosis
	Treatment Plan	1	1	treatment plan
Temporal tabular	Laboratory tests	73	71	serum triglyceride, serum creatinine
	Medications	17	21	angiotensin-converting enzyme inhibitor, heparin
	Vital signs measured	5	5	respiratory rate, pulse rate
Temporal note	Daily ward round records	1	1	daily ward round records

**Table 2 bioengineering-11-00094-t002:** Predictive performance of models with different main modality combinations.

Data Set	Selection of Main Modality	Prediction of In-Hospital Mortality			Prediction of Long Hospital Stay		
		AUROC	AUPRC	F1	AUROC	AUPRC	F1
AMI	Static notes and temporal notes	**0.928**	**0.363**	**0.535**	**0.881**	**0.632**	**0.478**
	Static tabular data and temporal notes	0.923	0.351	0.520	0.879	0.630	0.473
	Static notes and temporal tabular data	0.925	0.359	0.528	0.877	0.626	0.466
	Static tabular data and temporal tabular data	0.919	0.346	0.516	0.874	0.621	0.454
Stroke	Static notes and temporal notes	**0.954**	**0.455**	**0.671**	**0.847**	**0.508**	**0.376**
	Static tabular data and temporal notes	0.951	0.447	0.665	0.836	0.486	0.359
	Static notes and temporal tabular data	0.945	0.438	0.651	0.834	0.479	0.352
	Static tabular data and temporal tabular data	0.933	0.425	0.644	0.818	0.443	0.334

The bold in the table represents the optimal result.

**Table 3 bioengineering-11-00094-t003:** The results of evaluation experiments on the AMI dataset.

	Model	In-Hospital Mortality			Long Length of Stay		
		AUROC	AUPRC	F1	AUROC	AUPRC	F1
T-MAG	T-MAG	**0.928**	**0.363**	**0.535**	**0.881**	**0.632**	**0.478**
Neural Network	DNN	0.748	0.228	0.313	0.726	0.423	0.318
	LSTM	0.769	0.243	0.328	0.758	0.528	0.413
Fusion Methods	Fusion-CNN	0.816	0.267	0.403	0.716	0.513	0.405
	Fusion-LSTM	0.828	0.287	0.435	0.818	0.544	0.436
Fusion Methods	MulT	0.913	0.323	0.502	0.856	0.593	0.468
	Crossformer	0.893	0.319	0.491	0.855	0.597	0.461
	PatchTST	0.838	0.296	0.447	0.825	0.537	0.422
	MISTS-fusion	0.917	0.341	0.508	0.866	0.609	0.438
	Glaucoma-fusion	0.821	0.277	0.415	0.805	0.565	0.461
	MAG-DNN	0.844	0.312	0.481	0.838	0.557	0.445
	MAG-LSTM	0.916	0.339	0.511	0.874	0.619	0.451

The bold in the table represents the optimal result.

**Table 4 bioengineering-11-00094-t004:** The results of evaluation experiments on the Stroke dataset.

	Model	In-Hospital Mortality			Long Length of Stay		
		AUROC	AUPRC	F1	AUROC	AUPRC	F1
T-MAG	T-MAG	**0.954**	**0.455**	**0.671**	**0.847**	**0.508**	**0.376**
Neural Network	DNN	0.849	0.333	0.505	0.736	0.375	0.245
	LSTM	0.856	0.349	0.511	0.746	0.378	0.255
Fusion Methods	Fusion-CNN	0.879	0.388	0.573	0.767	0.408	0.286
	Fusion-LSTM	0.887	0.401	0.595	0.822	0.430	0.317
Fusion Methods	MulT	0.933	0.435	0.641	0.830	0.441	0.332
	Crossformer	0.945	0.450	0.658	0.844	0.498	0.370
	PatchTST	0.927	0.423	0.633	0.825	0.435	0.323
	MISTS-fusion	0.949	0.451	0.660	0.833	0.445	0.339
	Glaucoma-fusion	0.891	0.405	0.611	0.815	0.425	0.310
	MAG-DNN	0.914	0.411	0.625	0.799	0.419	0.301
	MAG-LSTM	0.938	0.445	0.647	0.836	0.488	0.358

The bold in the table represents the optimal result.

## Data Availability

Due to the protection of patient privacy, we are unable to disclose the raw data used.

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
