# Peer review of "Research on Multimodal Fusion of Temporal Electronic Medical Records"

_bioengineering, 2024, doi:10.3390/bioengineering11010094_

Round 1

Reviewer 1 Report

Comments and Suggestions for Authors

In this paper, the authors present Research on Multi-Modal Fusion of Temporal Electronic Medical Records. An attention-backtracking module was introduced for the latter, adept at capturing enduring dependencies in temporal fused representations. Concatenated results were channeled into an LSTM to yield the ultimate fused representation. Initially, two note modalities were designated as primary modes, and subsequently, the proposed fusion model was juxtaposed against baseline models. The proposed model consistently exhibited superior predictive prowess in both tasks. Removing the atten-tion-backtracking module led to performance decline. This study proffers a fresh outlook on the integration of temporal and static multimodal data, holding promise for broader applications in multimodal medical data representation. However, there are some issues should be addressed.

1. Due to the sparsity and varying lengths of medical temporal data, commonly used deep learning methods such as recurrent neural network (RNN) and LSTMs often struggle to effectively represent the temporal features of patients. How to solve this problem in the paper?

2. Common methods impose certain limits on the length of each input text. When dealing with texts that exceed a fixed length, they are often divided into multiple segments for separate processing. This not only leads to potentially incomplete semantics in the segmented parts but also makes it difficult to capture long-term dependencies. How to solve this problem?

3. The direct fusion of time-series and static representations can obscure the inherent temporal sequence information of the former. Does this paper use this concept?

4. In the initial application of MAG in the field of sentiment analysis, the choice of the primary modality is evident because text consistently contains the relatively richest emotional information. However, in the medical domain, this is harder to determine as each modality of data can provide its own unique information. How to overcome this problem?

5. With the widespread adoption of EMR systems, an increasing number of researchers are turning to deep learning methods for medical studies. However, current studies on multimodal EMR have certain limitations. How to break through this limitation?

Comments on the Quality of English Language

Minor editing of English language required

Reviewer 2 Report

Comments and Suggestions for Authors

Moderate editing of English language required

Comments on the Quality of English Language

Here are my comments/concerns about the manuscript:

  1. The abstract provides a clear overview of the research, detailing the methodology and outcomes. However, it lacks a concise mention of motivation and contributions, which should be addressed. Additionally, the abstract should reflect the comparison against recent models, not just baseline ones.

  1. The introduction lacks a clear motivation for the study. There should be a paragraph highlighting why a new solution is needed and the specific problems the proposed model aims to address. The discussion should emphasize the significance of integrating temporal and static multimodal data into electronic medical records. The current introduction focuses more on the prevalence of EMRs and less on why the proposed solution is necessary.

  2. The contributions of the work need to be explicitly outlined in the introduction. A separate paragraph should highlight the novel aspects introduced by the proposed multimodal fusion model, T-MAG.

  3. Add a paragraph at the end of the introduction summarizing the structure of the paper. This is standard practice in research papers and provides readers with an overview of what to expect.

  4. The discussion of previous work is informative, but it needs to go beyond listing strengths. Provide a more in-depth discussion of previous works related to the research topic. Present not only the strengths but also the limitations of each study. This will help in positioning the proposed work better in the context of existing research.

  5. The proposed solution needs a more systematic description. Provide a step-by-step algorithm for T-MAG to enhance reproducibility. The details are somewhat scattered, and a clearer structure would benefit readers attempting to implement the model. Include the necessary details so that others can implement the proposed solution in future studies.

  6. The discussion of experimental results is inadequate. Elaborate more on the implications of the findings. Why does the proposed model outperform baseline models? Discuss the practical significance of the results in the context of healthcare applications.

  7. The paper should extend its comparison beyond baseline models to include the most recent models in the field. This is crucial for establishing the superiority of the proposed model in the current research landscape.

  8. Clarify why the proposed work has only been compared against baseline models. Extend the comparison to the most recent models to establish the superiority of the proposed model. Justify why the baseline models were chosen and how they fit into the context of the study.

  9. The choice of LR and RF for comparison lacks justification. Provide a strong rationale for selecting these specific machine learning methods. Explain how LR and RF are suitable for comparison in the given study context.

  10. The quantity of experiments is insufficient for drawing solid conclusions. Increase the number of experiments and scenarios to ensure the robustness and reliability of the proposed model's performance.

  11. Discuss in more detail the impact of the attention backtracking module on the model's predictive performance. Provide insights into why this module is crucial and how it contributes to capturing long-term dependencies.

  12. The writing style is clear, but some sentences are too complex. Simplify the language for better readability.

  13. Ensure consistency in terminology and notation throughout the paper.

  14. Clearly define acronyms upon their first use.

  15. It's notable that the paper lacks recent references from 2023. Given the rapidly evolving nature of the field, it's crucial to include the latest studies and advancements to provide readers with the most up-to-date context. Incorporating recent references will enhance the credibility and relevance of the research.

  16. The caption for Figure 1 is extremely long. You should make it shorter. The current text in the caption can be placed in the text.

Reviewer 3 Report

Comments and Suggestions for Authors

The paper discusses a  model based on an integration of temporal and static multimodal data by using electronic records (main strength). The work is interesting but some weaknesses are found such as the discussion of the dataset and to reason of the choice of the machine learning model.  I suggest a minor revision by answering to the following points.

According to the weaknesses, I suggest to improve the paper by answering to these points:

·       Please provide more details about the analyzed dataset (number of records, attributes information and related statistics, etc.);     

·       Please discuss the pattern followed to fix the final hyperparameters of the adopted model;

·       Please provide a discussion about advantages, disadvantages, limitation and perspectives of the proposed model;

·       Authors should add more references about the application of machine learning in medical field, such as:

https://doi.org/10.3390/s23094178

doi: 10.1109/MeMeA49120.2020.9137224

https://doi.org/10.3390/healthcare10030541

https://doi.org/10.3390/app122211709

Finally, I suggest to revise the English language.

Comments on the Quality of English Language

To improve

Round 2

Reviewer 1 Report

Comments and Suggestions for Authors

The authors have solved the problems. It is good enough.

Comments on the Quality of English Language

Minor editing of English language required

Reviewer 2 Report

Comments and Suggestions for Authors

The following comments are yet to be addressed. According to you, all corrections have been made to the revised version. After comparing the original submitted draft with the revised draft. It is hard to see a significant difference between them.  

  1. Add a paragraph at the end of the introduction summarizing the structure of the paper. This is standard practice in research papers and provides readers with an overview of what to expect.

  2. The discussion of previous work is informative, but it needs to go beyond listing strengths. Provide a more in-depth discussion of previous works related to the research topic. Present not only the strengths but also the limitations of each study. This will help in positioning the proposed work better in the context of existing research.

  3. The proposed solution needs a more systematic description. Provide a step-by-step algorithm for T-MAG to enhance reproducibility. The details are somewhat scattered, and a clearer structure would benefit readers attempting to implement the model. Include the necessary details so that others can implement the proposed solution in future studies.

  4. The discussion of experimental results is inadequate. Elaborate more on the implications of the findings. Why does the proposed model outperform baseline models? Discuss the practical significance of the results in the context of healthcare applications.

  5. The paper should extend its comparison beyond baseline models to include the most recent models in the field. This is crucial for establishing the superiority of the proposed model in the current research landscape.

  6. Clarify why the proposed work has only been compared against baseline models. Extend the comparison to the most recent models to establish the superiority of the proposed model. Justify why the baseline models were chosen and how they fit into the context of the study.

  7. The choice of LR and RF for comparison lacks justification. Provide a strong rationale for selecting these specific machine learning methods. Explain how LR and RF are suitable for comparison in the given study context.

  8. The quantity of experiments is insufficient for drawing solid conclusions. Increase the number of experiments and scenarios to ensure the robustness and reliability of the proposed model's performance.

  9. Discuss in more detail the impact of the attention backtracking module on the model's predictive performance. Provide insights into why this module is crucial and how it contributes to capturing long-term dependencies.

  10. The caption for Figures 1 and 3 is extremely long. You should make it shorter. The current text in the caption can be placed in the text.

Comments on the Quality of English Language

Moderate editing of English language required

Author Response

请参阅附件

Round 3

Reviewer 2 Report

Comments and Suggestions for Authors

One more correction is needed.

The remainder of this paper is organized as follows. Section 2 first introduces the..... 

The above paragraph must be placed at the end of the introduction section. It should be placed after the (The contributions of this paper can be summarized as follows:) paragraph!

Comments on the Quality of English Language

Moderate editing of English language required

Author Response

Reviewer #: One more correction is needed.

The remainder of this paper is organized as follows. Section 2 first introduces the..... 

The above paragraph must be placed at the end of the introduction section. It should be placed after the (The contributions of this paper can be summarized as follows:) paragraph!

Response: Thank you for your suggestion. As you suggested, we have moved this paragraph to the end of the introduction, to lines 136-141. Hope this modification can meet your requirements.